# Associations between Milk Coagulation Properties and Microbiological Quality in Sheep Bulk Tank Milk

**DOI:** 10.3390/foods13060886

**Published:** 2024-03-14

**Authors:** Ramón Arias, Lorena Jiménez, Ana Garzón, Javier Caballero-Villalobos, Bonastre Oliete, Nicolò Amalfitano, Alessio Cecchinato, José M. Perea

**Affiliations:** 1Instituto Regional de Investigación y Desarrollo Agroalimentario y Forestal de Castilla-La Mancha (IRIAF), CERSYRA de Valdepeñas, 13300 Ciudad Real, Spain; rarias@jccm.es; 2Departamento de Producción Animal, Universidad de Córdoba, 14071 Córdoba, Spain; pa1gasia@uco.es (A.G.); javier.caballero@uco.es (J.C.-V.); jmperea@uco.es (J.M.P.); 3Institut Agro, Université Bourgogne Franche-Comté, PAM UMR A 02.102, 21000 Dijon, France; bonastre.oliete@u-bourgogne.fr; 4Department of Agronomy, Food, Natural Resources, Animals and Environment (DAFNAE), University of Padova, Viale dell’Università 16, Legnaro, 35020 Padova, Italy; nicolo.amalfitano@unipd.it (N.A.); alessio.cecchinato@unipd.it (A.C.)

**Keywords:** coagulation traits, dairy industry, microbiological quality, milk sheep quality, tank milk

## Abstract

This study conducted a seasonal analysis of bulk tank milk from 77 sheep farms to establish relationships between the concentration of major microbial groups and milk coagulation properties. The investigated milk traits included composition (pH, fat, casein, lactose), coagulation properties (curd firmness: A_60_-, rennet clotting time: RCT-, curd firming time: k_20_-, curd yield: CY-), and somatic cell score (SCS). The main microbial groups analyzed were total mesophilic bacteria (SPC), thermodurics (THERMO), psychrotrophs (PSYCHRO), *Pseudomonas* spp. (PSEUDO), lactic acid bacteria (LAB), catalase-negative gram-positive cocci (GPCNC), *Escherichia coli* (ECOLI), coliforms other than *Escherichia coli* (COLI), coagulase-positive staphylococci (CPS), coagulase-negative staphylococci (CNS), and spores of lactate-fermenting *Clostridium* (BAB). Mixed linear models were used to explore associations between coagulation properties and the aforementioned variables. Results demonstrated that incorporating microbial loads into the models improves their fit and the relative quality of the outcomes. An important seasonality is demonstrated by an increase in CY and A_60_, along with a decrease in RCT and k_20_ during autumn and winter, contrasting with spring and summer. BAB concentration resulted in a reduction of A_60_ and an increase in RCT, whereas SPC concentration led to an enhancement of A_60_ and a reduction in RCT. An increase in GPCNC concentration was associated with an increase in k_20_ and a decrease in CY.

## 1. Introduction

Sheep milk production holds significant importance in countries across the Mediterranean basin. Traditionally, native breeds have been raised, with many now facing the threat of extinction. These breeds have played a crucial role in sustaining livestock activity and associated industries in vast rural areas. This not only helps to maintain local populations but also contributes to environmental protection [1]. Sheep milk production is primarily directed towards cheese and dairy product processing, with many of these products holding protected designation of origin (PDO) status [2]. Many of these cheeses are known for their unique sensory characteristics, which result from traditional manufacturing techniques. The quality of sheep milk is a key factor in creating exceptional cheeses and is subject to rigorous quality control systems in the industry [3]. Traditionally, milk quality assessment has focused on physicochemical aspects, mainly fat and protein content. Recently, hygienic parameters have been integrated into these quality systems, including analyses like total mesophilic microorganism and somatic cell counts (SCC). However, practices considering aspects such as differential milk microbiology are still limited [3].

Research in dairy cattle has provided valuable insights into the microbial composition of milk and its relationship with several sources of contamination, emphasizing the importance of this aspect in milk quality [4]. These contaminants introduce a variety of microorganisms associated with farm environments, such as lactic acid bacteria, hygiene practices on farms leading to the presence of coliforms, and intramammary infections caused by staphylococci [5]. Other researchers [6,7] have also highlighted the complex microbial composition found in milk from different species, emphasizing the importance of its contamination sources and its impact on cheese-making processes, especially concerning lactic acid bacteria. Some studies have even suggested that stringent hygiene measures have significantly reduced microbial populations, potentially adversely affecting the production of traditional and artisanal cheeses [8]. However, there is limited research specifically addressing the differential microbial load and microbiota of sheep milk [9,10].

Several studies have underscored the significance of milk coagulation properties (MCP) as complementary traits alongside the inherent quality of raw milk [11,12]. For that reason, there is well-documented literature on the relevance and significance of MCP, mainly in cattle [13,14] and, to a lesser extent, in sheep and goats [15,16]. Also, these studies underline the importance of improving milk MCP and explore the relationship with several factors that contribute its modification, such as breeding, feeding, farm management system, season, or milking system. Factors such as bacterial counts and the intricate interplay among microbiological characteristics and technological parameters can influence MCP and cheese production. However, research in this area, particularly regarding its impact on milk from small ruminants, remains limited [5].

Therefore, this study aims to assess the impact of composition and microorganisms present in sheep milk on its coagulation properties. Additionally, it seeks to compare the predictive performance of models incorporating microbiological variables with reference models that do not include these variables. The ultimate goal is to reach a better understanding of the microbiological factors influencing milk coagulation, providing valuable information for quality control and dairy production management.

## 2. Materials and Methods

### 2.1. Study Design and Sample Determination

The present study included 77 Manchega dairy sheep flocks from the region of Castilla-La Mancha (Spain), registered under Protected Designation of Origin “Queso Manchego”, representing 15% of the farms listed in PDO [17]. The average flock size was 1144 ewes, ranging from 150 to 5500 ewes, with an average production of 143 kg/ewe/year, varying from 91 to 200 kg, over a mean lactation period of 121 days, ranging from 100 to 150 days. All farms employ mechanical milking systems, and ewes are milked twice daily. The milk is then filtered and stored in refrigerated tanks, maintaining a maximum temperature of 6 °C. The production system is semi-intensive, traditionally associated with grazing on natural pastures and utilizing the residues and remains of cereal crops. On average, each sheep has access to 1.44 ha, ranging from 0.5 to 1.94 ha.

The flocks were visited four times a year (once during each season) to collect bulk tank samples, which included milk from the morning milking of the visit day and the previous evening milking. Samples were stored in hermetically sealed sterilized containers and kept at 4 °C until the arrival at the Experimental Dairy Science and Cheese Laboratory at the Regional Centre for Animal Selection and Reproduction (CERSYRA-IRIAF, Valdepeñas, Ciudad Real, Spain), where they were aliquoted and sent to other facilities for further analyses.

### 2.2. Laboratory Analysis

Microbiological analyses were conducted at CERSYRA-IRIAF within the first 24 h following sample collection. The techniques and methods used to determine the microbiological counts are described in detail in a previous paper from our research group [5]. Serial dilutions were performed from each sample to inoculate 0.1 mL onto different culture media. Bacterial counts for the following groups of microorganisms were determined on PCA media (Panreac, Barcelona, Barcelon, Spain): standard plate count (SPC) was incubated in aerobic conditions at 30 °C for 72 h [18]; thermoduric bacteria (THERMO, Panreac, Barcelona, Spain) were incubated under the same conditions as SPC, after pasteurizing milk at 62.8 °C for 30 min [19]; and psychrotrophic bacteria (PSYCHRO, Panreac, Barcelona, Spain) were incubated at 6.5 °C for 10 days [20]. *Pseudomonas* spp. (PSEUDO, Panreac, Barcelona, Spain) was cultured on Cetrimide agar (Panreac, Spain) and incubated at 35 °C for 48 h [21]. Determination of *Escherichia coli* (ECOLI, Basingstoke Hampshire, UK) and other coliforms (COLI, Basingstoke Hampshire, UK) was conducted using CromoIDTM Coli medium (bioMérieux, Madrid, Spain), incubated at 37 °C for 24 h [22]. Gram-positive catalase-negative cocci count (GPCNC) was determined in modified Edwards medium with colistin and oxolinic acid supplement (Oxoid, Basingstoke Hampshire, UK), incubated at 35 °C for 48 h [9]. Lactic acid bacteria (LAB) were seeded on MRS medium (Panreac, Barcelona, Spain) acidified to pH 5.7, and incubated at 30 °C for 72 h [23]. Lactate-fermenting Clostridium spore (BAB, Basingstoke Hampshire, UK) count was performed using the most probable number (MPN) technique in Bryant and Burkey Broth (BBB, Merck, Darmstadt, Germany) [24]. For the enumeration of coagulase-positive staphylococci (CPS) and coagulase-negative staphylococci (CNS), Baird–Parker RPF agar medium (bioMérieux, Madrid, Spain) was used, and incubated at 37 °C for 24 h [4]. The microbial counts were expressed in colony-forming units per mL of milk (CFU/mL), and Clostridium spores were expressed as spores/mL. A logarithmic transformation (log_10_) was applied to both microbial counts to normalize their distribution.

Milk composition, pH, and somatic cell count (SCC) analyses were conducted at the Interprofessional Dairy Laboratory of Castilla-La Mancha (LILCAM, Talavera de la Reina, Toledo, Spain) within 48 h from sample collection. The pH was measured using a Crison Basic 20 pH meter (Crison Instruments, Barcelona, Spain). Milk composition, including fat content (FAT), total casein (CAS), and lactose (LAC) was determined using a Milkoscan 6000 FT device (Foss Electric, Hillerød, Denmark), which operates on mid-infrared spectroscopy. The total SCC was quantified by flow cytometry on a Fossomatic FC (Foss Electric, Hillerød, Denmark). The SCC was subsequently expressed as somatic cell scores (SCS) by applying the following formula [25]:SCS = log_2_[SCC cells/mL × 10^−5^] + 3

Milk coagulation properties were monitored at the Small Ruminant Dairy Laboratory (University of Córdoba, Spain) within the first 24 h following sample collection. Using a Formagraph viscosimeter (Foss Electric A/S, Hillerød, Denmark), key coagulation parameters were determined based on the methodology adapted from McMahon and Brown [26]. These parameters included rennet coagulation time (RCT), curd-firming time (k_20_), curd firmness at 60 min after rennet addition (A_60_), and curd yield (CY). Curd yield was subsequently calculated following draining of the fresh curds [5].

### 2.3. Statistical Analysis

Data analysis was performed on SAS version 9.3 (SAS Institute Inc., Cary, NC, USA). Preliminary testing of the data was carried out to determine outliers to be discarded before further analysis. After the editing, a total of 306 data points were used (Table 1). The explanatory variables (i.e., SCS, BAB, SPC, THERMO, PSYCHRO, PSEUDO, LAB, GPCNC, ECOLI, COLI, CPS, CNS) were discretized into 5 classes based on the 20th, 40th, 60th, and 80th percentiles. This approach was taken to ensure a better assessment of the pattern of microbiological count effects and to guarantee the classes were balanced [27].

A linear mixed model (MIXED) procedure was used to investigate the associations between microbiological quality and milk coagulation properties. The model is represented as follows:$$Y=\sum_{i=1}^{n}\beta_{i}f_{i}+\delta$$
where *Y* denotes milk coagulation traits (CY, A_60_, RCT, k_20_), *β_i_* are the unknown parameters to be estimated, *f_i_* denotes the explanatory variables, and *δ* is the error term.

Two models were developed for each dependent variable. Initially, a baseline model (BM) was established, excluding microbiological counts, with the purpose of comparison against a second model (whole set model: WSM) that included all relevant variables. The BM model was adjusted, considering flock as a random factor (77 classes), while season (4 classes: spring, summer, autumn, winter) and SCS (5 classes: 4.44–5.72; 5.72–6.23, 6.23–6.65, 6.65–7.01, 7.01–8.90 log_2_ SCC) were set up as fixed effects. Additionally, covariates such as pH, FAT (%), CAS (%), and LAC (%) were incorporated.

The WSM model incorporating microbiological counts was adjusted using the same variables as BM, completing it with microbiological counts discretized into five classes as fixed factors (Table 2).

To address challenges arising from the limited sample size (308 complete cases), a step-by-step sequence was undertaken, considering the primary effects of all fixed, random, and covariate factors. A significance level of *p* < 0.05 was used for the retention criterion. In the initial phase, all conceivable single-variable models were assessed using the Akaike Information Criterion (AIC) value. Subsequently, each remaining factor was incrementally added, one by one, to the model with the lowest AIC value, and each predictor was compared based on the AIC value. This iterative process continued until the model with the lowest AIC was identified [28]. This model was deemed the most plausible and was selected as the final model.

After defining the model, an analysis of collinearity was conducted using the variance inflation factor (VIF) of the regression coefficients. A severe multicollinearity issue was present if any of the VIFs exceeded 10 [28]. The normal distribution of residuals was verified using Kolmogorov–Smirnov test, while the absence of autocorrelation of residuals was assessed through the Durbin–Watson test. Heteroscedasticity was evaluated using White’s test. The model fit was assessed using the adjusted coefficient of determination (R^2^) and the root mean squared error (RMSE) [29].

Polynomial contrasts (*p* < 0.05) were computed to delineate the pattern of effects of microbiological counts (BAB, SPC, THERMO, PSYCHRO, PSEUDO, LAB, GPCNC, ECOLI, COLI, CPS, CNS). First-order comparisons measure linear relationships, while second-order comparisons measure quadratic relationships.

## 3. Results and Discussion

The most plausible models identified for each of the analyzed coagulation characteristics are shown in Table 3. Broadly, the models exhibited poor fit (R^2^ < 50%), except for CY. In the case of curd yield, the coefficient of determination ranged between 54.8% for the baseline model and 57.9% for the model incorporating microbiological counts. This emphasizes the fact that the inclusion of microbiological counts not only significantly improved the model fit but also enhanced relative quality across all evaluated outcomes.

A significant association was observed between season and all coagulation traits. This association persisted even in models that incorporated microbiological counts, albeit with a slightly reduced contribution. As expected, season was a significant source of variation for all investigated traits, underscoring the importance of considering this effect in the statistical model for a better evaluation of the effect of microbiological counts. It is important to note that Manchega does not exhibit as pronounced a seasonality in milk production as other Mediterranean breeds, such as Sarda [30] or Latxa [31]. Therefore, the seasonal differences found are arguably more interesting in terms of their relevance to the cheese industry.

The impact of flock on the variability of the examined coagulation traits was modest, as indicated by previous research [11,32]. Although the AIC favored models excluding the flock effect, this does not completely rule out its influence. These results are explained by the fact that the evaluated flocks follow a similar farming system, with other variables within the scope of this study exerting a more pronounced influence on coagulation characteristics than flock variability.

The level of somatic cells was only significantly associated with RCT. This result is in line with a previous study by Pazzola et al. [33] focused on the Sarda breed, in which higher somatic cell counts were linked to slower coagulations. Moreover, Caballero-Villalobos et al. [34] observed in individual samples from Manchega that an increase in SCC led to a rise in RCT and k_20_ and a decrease in curd firmness measured at 30 min. This could be attributed to an increase in casein breakdown as a consequence of increased plasmin activity, the main enzyme associated with somatic cells in milk [35].

Regarding CY, the most plausible baseline model was adjusted considering season, CAS, FAT, and pH, in that order of importance. The potential inclusion of microbiological counts led to a better model, which specifically incorporated the level of gram-positive catalase-negative cocci (GPCNC), thus excluding pH. As expected, CY was positively associated with FAT and CAS, which is consistent with the correlations found for these quality parameters and CY in Manchega sheep milk [5]. In other breeds, such as Sarda, Assaf, or Churra, this relationship has also been documented [30,36,37]. CY was also negatively associated with the level of GPCNC, with the linear pattern of the relationship being confirmed across the classes of GPCNC (Figure 1).

It has been described that high microbial counts could be associated with an enzymatic load acting proteolytically and lipolytically on the milk, leading to a decrease in its fat and protein content, indirectly affecting coagulation properties [38]. In terms of season, CY was higher in autumn and winter milk than in spring and summer. The higher yields obtained in these seasons could be related to the increased levels of the major components of the milk, as has also been shown in various studies [39,40].

Regarding A_60_, the best baseline model was adjusted with season and CAS, the latter being the most important factor. However, the model fit was the poorest of all, with a coefficient of determination of 8.5%. The possibility of including microbiological counts led to improving the model, incorporating BAB and SPC, and also including LAC. The level of BAB acquired a similar importance to CAS for A_60_. The association of A_60_ was positive with CAS and SPC and negative with LAC and BAB (Figure 2). SPC and BAB levels showed a good fit with linear and quadratic patterns. The associations found are consistent with the correlations for these quality parameters and A_60_ in Manchega [5] and Sarda [11,12] sheep milk, which indicate that the decrease in curd firmness is inversely proportional to the increase in lactose content, with the relationship between firmer curds and the level of CAS being noteworthy. An opposite effect of the vegetative forms of total mesophilic microorganisms (SPC) and spore-forming forms in milk (BAB) on curd hardness was observed, for which no previous references have been found, and which will be appropriate to address in future studies. A_60_ follows a seasonal pattern similar to CY, with increased curd firmness in autumn and winter. Todaro et al. [41] found greater hardness in spring and autumn, possibly related to the production system itself, which conditions milk quality.

The most plausible baseline model for RCT was adjusted with LAC, season, and somatic cell level, achieving a coefficient of determination of 39.1%. LAC and season were the most relevant factors for the onset of coagulation. In contrast, the model that incorporated microbiological levels was more plausible, obtaining a coefficient of determination of 44.4%. This model included the levels of SPC and BAB and also incorporated pH, this time excluding somatic cell levels. The association of RCT was positive with pH, LAC, and BAB and negative with the SPC. These results are in line with those reported by other authors [11,12], who reported an increase in coagulation times when individual sheep milk samples have a more alkaline pH, although this same study shows a negative relationship between lactose content and RCT. The mechanisms that exactly explain the association between lactose and technological traits are still unknown, although it could be associated with a modification of the percentage and composition of minerals and proteins, in accordance with the role of lactose as an osmotic regulator of milk [11,12]. BAB followed a linear pattern, while linear and quadratic patterns were confirmed for SPC (Figure 3). It is confirmed that different types of microorganisms do not have the same effect on RCT, as was also observed for curd hardness; on one hand, active vegetative cells, such as SPC, consume lactose to assure growth and, consequently, decrease RCT, while spore-forming microorganisms like BAB are associated with a longer duration of coagulation, which would have certain negative connotations in optimizing the cheese-making process. Regarding season, RCT is significantly higher in spring and lower in autumn, similar to results obtained in Latxa sheep, with increased RCT from April to July [31], and in dairy cattle, with the lower value in autumn [42]. Conversely, other studies reported the lowest coagulation times in spring and the highest in winter for bulk tank sheep milk [41]. In any case, in our study, differences in coagulation time depending on season are much higher than those of the cited studies, which, as previously mentioned, may be related to the production system of this particular breed.

The best baseline model for k_20_ was adjusted with season, CAS, and pH, achieving a poor coefficient of determination of 10.4%. The model with microbiological counts added the GPCNC level and increased the coefficient of determination to 13.1%. The association of k_20_ was positively associated with pH, similar to what was observed for RCT, and negatively associated with CAS, similar to GPCNC, which showed a negative linear pattern (Figure 4). The relationship between k_20_ and CAS follows the trend already highlighted in previous studies, particularly when coagulation times are high (RCT > 30 min) [5]. According to some authors, high microorganism counts, as in this case, GPCNC, affect coagulation by causing acidification of the milk and a faster firming of the curd [38]. Similar to RCT, k_20_ was significantly higher in spring and lower in autumn. Others, on the contrary, reported lower levels for k_20_ in spring and winter [41], differences that could be due, as discussed for other parameters, to either the production system of Valle del Belice sheep, which is more extensive and dependent on the availability of pastures, or to differences in the stage of lactation [43,44].

## 4. Conclusions

The applied models demonstrated limited fitting for most milk coagulation properties, except for curd yield (CY). Incorporating microbiological analysis into the models significantly enhanced both the fit and the relative quality of the results, highlighting the importance of microbiological factors in dairy industry quality control systems. Significant associations were found between seasonal variations and all coagulation properties, even when microbiological counts were considered. During autumn and winter, the coagulation process initiates earlier (RCT) and proceeds at a slower pace (k_20_), resulting in firmer curds (A_60_) and yielding better curd production (CY). Both lactate-fermenting Clostridium spores (BAB) and total mesophilic bacteria (SPC) showed significant associations with A_60_ and RCT, albeit with opposing effects. While BAB concentration reduced curd firmness and increased coagulation onset, SPC concentration enhanced curd firmness and promoted coagulation onset. Coagulation speed and curd yield were significantly associated with gram-positive catalase-negative cocci (GPCNC). As GPCNC concentration increased, coagulation speed accelerated, and curd yield decreased. These results emphasize the complexity of interactions among microbiological factors and milk composition from farms and their collective impact on coagulation characteristics and seasonal variations, a crucial aspect for optimizing cheese production.

## Figures and Tables

**Figure 1 foods-13-00886-f001:**
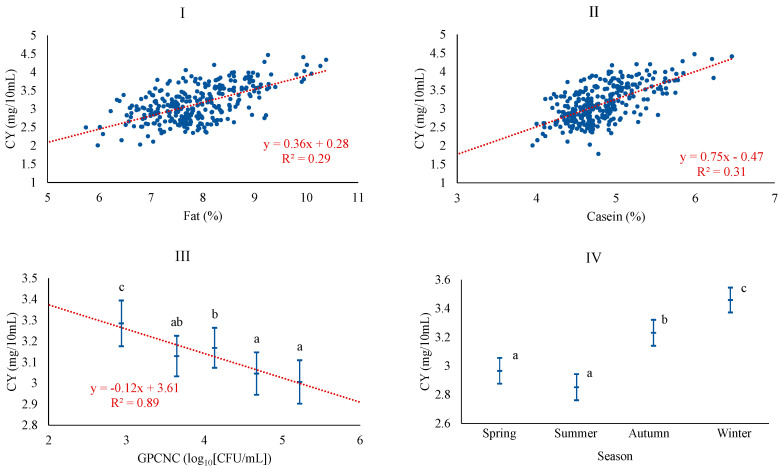
Least squares means and 95% CI of curd yield (CY) across significant variables (I: FAT (%); II: CAS (%); III: GPCNC, gram-positive catalase-negative cocci (log_10_[CFU/mL]); IV: season). When the trend line is presented, it means that the linear contrast (₋ ₋ ₋) was significant at *p* < 0.05. Means with different superscript letters are different at *p* < 0.05 (SNK test).

**Figure 2 foods-13-00886-f002:**
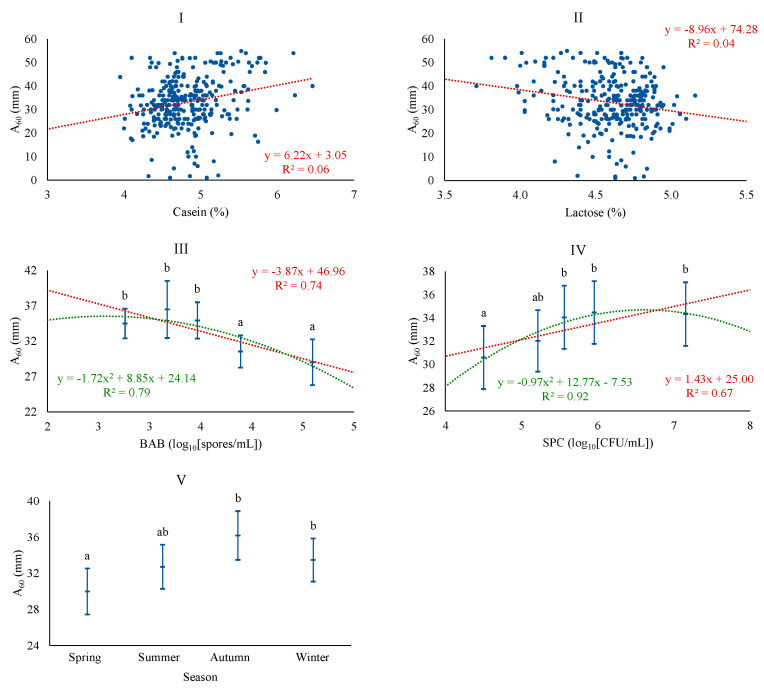
Least squares means and 95% CI of maximum curd firmness (A_60_) across significant variables (I: CAS (%); II: LAC (%); III: BAB, butyric acid bacteria (log_10_[spores/mL]); IV: SPC, total mesophilic bacteria (standard plate count) (log_10_[CFU/mL]); V: season). When the trend line is presented, it means that the linear (₋ ₋ ₋) and/or quadratic (₋ ₋ ₋) contrast was significant at *p* < 0.05. Means with different superscript letters are different at *p* < 0.05 (SNK test).

**Figure 3 foods-13-00886-f003:**
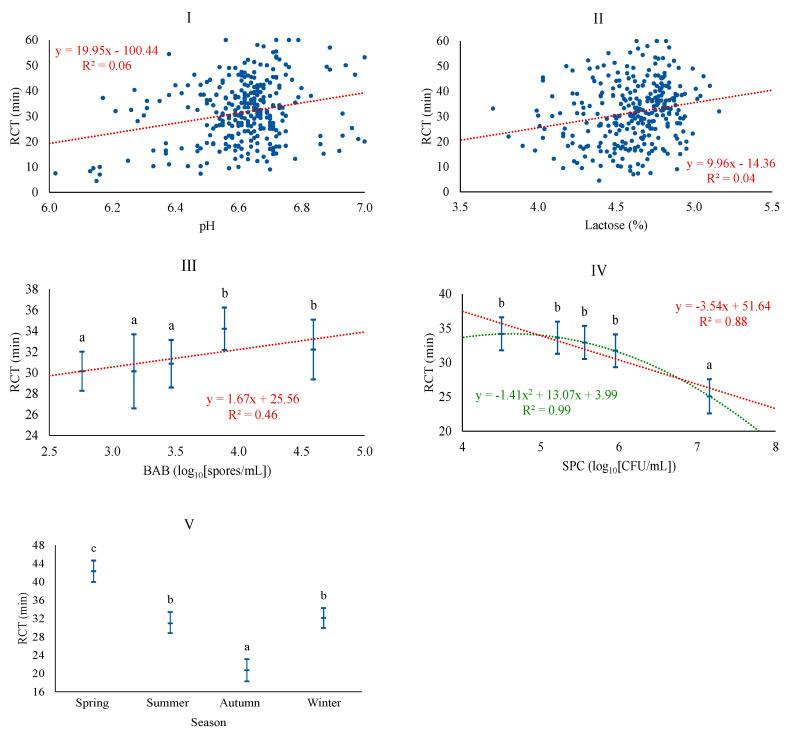
Least squares means and 95% CI of rennet clotting time (RCT) across significant variables (I: pH; II: LAC (%); III: BAB, butyric acid bacteria (log_10_[spores/mL]); IV: SPC, total mesophilic bacteria (standard plate count) (log_10_[CFU/mL]); V: Season). When the trend line is presented, it means that the linear (₋ ₋ ₋) and/or quadratic (₋ ₋ ₋) contrast was significant at *p* < 0.05. Means with different superscript letters are different at *p* < 0.05 (SNK test).

**Figure 4 foods-13-00886-f004:**
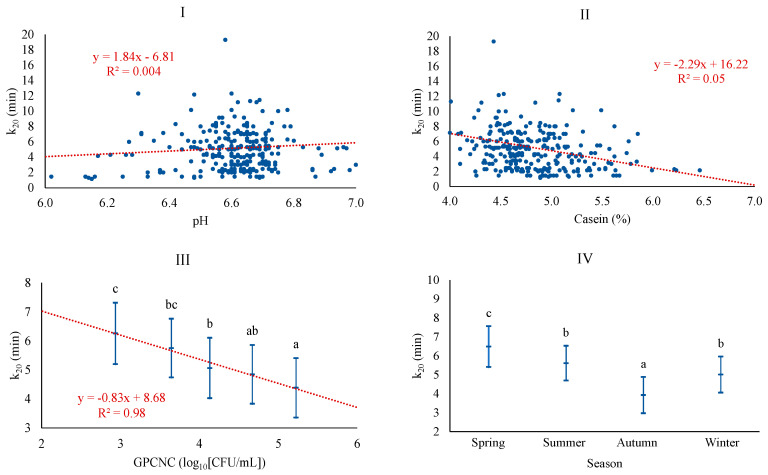
Least squares means and 95% CI of curd-firming time (k_20_) across significant variables (I: pH; II: CAS (%); III: GPCNC, gram-positive catalase-negative cocci (log_10_[CFU/mL]); IV: season). When the trend line is presented, it means that the linear contrast (₋ ₋ ₋) was significant at *p* < 0.05. Means with different superscript letters are different at *p* < 0.05 (SNK test).

**Table 1 foods-13-00886-t001:** Descriptive statistics of the variables for composition, microbiological quality, and coagulation properties of sheep milk from bulk tank (*n* = 308).

Variable	Description	Mean	SD ^1^	Q_1_	Q_3_
CY (g/100 mL) ^2^	Curd yield	3.13	0.55	2.76	3.74
A_60_ (mm) ^3^	Curd firmness at 60 min.	33.04	10.76	26.30	39.98
RCT (min) ^4^	Rennet clotting time	31.54	11.97	22.30	39.45
k_20_ (min) ^5^	Curd firming time	8.79	14.17	3.0	7.0
SCS ^6^	Somatic cell score	6.40	0.78	5.91	6.90
FAT (%)	Fat content	7.87	0.83	7.32	8.35
CAS (%)	Casein content	4.82	0.41	4.55	5.04
LAC (%)	Lactose content	4.60	0.24	4.48	4.79
pH (−log [H+])	pH	6.61	0.15	6.56	6.69
BAB ^7^	Lactate-fermenting *Clostridium* spores	3.42	0.63	2.96	3.80
SPC ^8^	Total mesophilic bacteria (standard plate count)	5.65	0.77	5.11	6.13
THERMO ^8^	Thermodurics	3.25	0.84	2.69	3.73
PSYCHRO ^8^	Psychrotrophs	5.23	1.08	4.57	5.90
PSEUDO ^8^	*Pseudomonas* spp.	3.15	0.84	2.65	3.72
LAB ^8^	Lactic acid bacteria	4.89	0.73	4.30	5.40
GPCNC ^8^	Gram-positive catalase-negative cocci	4.16	1.63	3.55	4.78
ECOLI ^8^	*Escherichia coli*	1.63	1.16	0.13	2.31
COLI ^8^	Coliforms other than *Escherichia coli*	2.91	1.19	2.35	3.67
CPS ^8^	Coagulase-positive staphylococci	2.52	1.35	1.90	3.46
CNS ^8^	Coagulase-negative staphylococci	4.30	0.54	4.06	4.50

^1^ SD: standard deviation, Q_1_: 1st quartile; Q_3_: 3rd quartile. ^2^ CY: curd yield. ^3^ A_60_: curd firmness at 60 min. ^4^ RCT: rennet clotting time. ^5^ k_20_: curd firming time. ^6^ SCS: bulk tank milk somatic cells score (log_2_[SCC cells/mL × 10^−5^] + 3). ^7^ BAB: lactate-fermenting *Clostridium* spores (log_10_[spores/mL]). ^8^ SPC: total mesophilic bacteria (standard plate count), THERMO: thermodurics, PSYCHRO: psychrotrophs, PSEUDO: *Pseudomonas* spp., LAB: lactic acid bacteria, GPCNC: gram-positive catalase-negative cocci, ECOLI: *Escherichia coli*, COLI: coliforms other than *Escherichia coli*, CPS: coagulase-positive staphylococci, CNS: coagulase-negative staphylococci. All microbial counts are expressed as: log_10_(UFC/mL).

**Table 2 foods-13-00886-t002:** Classes of microbial counts used in linear mixed models as fixed factors (*n* = 308).

Variable	Classes				
	1	2	3	4	5
BAB ^1^	4.04–4.85	4.85–5.26	5.26–5.38	5.38–5.89	5.89–7.00
SPC ^2^	2.48–3.04	3.04–3.30	3.30–3.63	3.63–4.15	4.15–5.04
THERMO ^2^	3.99–5.01	5.01–5.41	5.41–5.70	5.70–6.20	6.20–8.10
PSYCHRO ^2^	0.00–2.60	2.60–3.03	3.03–3.30	3.30–3.89	3.89–5.48
PSEUDO ^2^	2.00–4.43	4.43–4.98	4.98–5.36	5.36–6.14	6.14–8.00
LAB ^2^	0.00–2.55	2.55–2.95	2.95–3.34	3.34–3.80	3.80–4.53
GPCNC ^2^	2.91–4.22	4.22–4.66	4.66–5.11	5.11–5.46	5.46–6.48
ECOLI ^2^	0.00–2.48	2.48–3.39	3.39–4.37	4.37–4.97	4.97–5.48
COLI ^2^	0.00–1.00	1.00–1.65	1.65–1.96	1.96–2.48	2.48–5.48
CPS ^2^	0.00–2.45	2.45–2.89	2.89–3.31	3.31–3.98	3.98–5.48
CNS ^2^	0.00–1.65	1.65–2.50	2.50–3.06	3.06–3.59	3.59–5.48

^1^ BAB: lactate-fermenting Clostridium spores (log_10_[spores/mL]). ^2^ SPC: total mesophilic bacteria (standard plate count), THERMO: thermodurics, PSYCHRO: psychrotrophs, PSEUDO: *Pseudomonas* spp., LAB: lactic acid bacteria, GPCNC: gram-positive catalase-negative cocci, ECOLI: *Escherichia coli*, COLI: coliforms other than *Escherichia coli*, CPS: coagulase-positive staphylococci, CNS: coagulase-negative staphylococci. All microbial counts are expressed as: log_10_(UFC/mL).

**Table 3 foods-13-00886-t003:** More plausible linear mixed models (F-value and significance) based on the composition of the milk (BM), and the incorporation of microorganism counts (WSM) (*n* = 308).

Variable ^1^	CY (g/100 mL)		A_60_ (mm)		RCT (min)		k_20_ (min)	
	BM	WSM	BM	WSM	BM	WSM	BM	WSM
Flock	ns	ns	ns	ns	ns	ns	ns	ns
Season	31.27 ***	26.47 ***	4.35 ***	2.55 *	51.99 ***	35.56 ***	5.87 ***	4.72 **
SCS ^2^	ns	ns	ns	ns	2.27 *	ns	ns	ns
FAT (%)	21.71 ***	13.30 ***	ns	ns	ns	ns	ns	ns
CAS (%)	26.07 ***	14.71 ***	7.60 ***	6.98 ***	ns	ns	5.77 **	5.10 *
LAC (%)	ns	ns	ns	3.75 *	70.75 ***	12.81 ***	ns	ns
pH (−log[H+])	2.76 *	ns	ns	ns	ns	55.69 ***	8.51 ***	5.68 *
BAB ^3^	-	ns	-	6.20 ***	-	4.44 ***	-	ns
SPC ^4^	-	ns	-	3.04 **	-	6.25 ***	-	ns
THERMO ^4^	-	ns	-	ns	-	ns	-	ns
PSYCHRO ^4^	-	ns	-	ns	-	ns	-	ns
PSEUDO ^4^	-	ns	-	ns	-	ns	-	ns
LAB ^4^	-	ns	-	ns	-	ns	-	ns
GPCNC ^4^	-	3.28 **	-	ns	-	ns	-	2.97 *
ECOLI ^4^	-	ns	-	ns	-	ns	-	ns
COLI ^4^	-	ns	-	ns	-	ns	-	ns
CPS ^4^	-	ns	-	ns	-	ns	-	ns
CNS ^4^	-	ns	-	ns	-	ns	-	ns
AIC ^5^	−592.06	−445.62	1409.53	1116.48	1356.69	1067.44	754.21	634.08
RMSE ^6^	0.37	0.33	10.31	9.76	9.38	8.82	3.77	3.03
R^2 7^	54.80	57.9	8.50	13.80	39.1	44.44	10.40	13.10

^1^ CY: curd yield, A_60_: curd firmness at 60 min., RCT: rennet clotting time, k_20_: curd firming time. BM: baseline model, WM: whole set of variables model. ^2^ SCS: bulk tank milk somatic cells score (log_2_[SCC cells/mL × 10^−5^] + 3). ^3^ BAB: lactate-fermenting *Clostridium* spores (log_10_[spores/mL]). ^4^ SPC: total mesophilic bacteria (standard plate count), THERMO: thermodurics, PSYCHRO: psychrotrophs, PSEUDO: *Pseudomonas* spp., LAB: lactic acid bacteria, GPCNC: gram-positive catalase-negative cocci, ECOLI: *Escherichia coli*, COLI: coliforms other than *Escherichia coli*, CPS: coagulase-positive staphylococci, CNS: coagulase-negative staphylococci. All microbial counts are expressed as: log_10_(UFC/mL). ^5^ AIC: Akaike information criterion. ^6^ RMSE: root mean squared error. ^7^ R^2^: Determination coefficient. *: *p* < 0.05, **: *p* < 0.01, *** *p* < 0.001, ns: *p* > 0.05.

## Data Availability

The original contributions presented in the study are included in the article, further inquiries can be directed to the corresponding author.

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
