# Peer review of "Associations between Milk Coagulation Properties and Microbiological Quality in Sheep Bulk Tank Milk"

_foods, 2024, doi:10.3390/foods13060886_

Round 1
Reviewer 1 Report
Comments and Suggestions for Authors
Very good article, the topic is important, and the results and developed models are valuable and allow to explore the topic of coagulation and the impact of the presence of microorganisms on this process. Some issues should be improved:
- - Literature gap is weakly stated. Please develop introduction.
- - Line 37: …on the production of high-quality…
- - Line 50-51: Please redraft the sentence about the coagulation process
- - How many sheep were on the farms, please provide information about the flocks size and milk yield in the materials and methods section
- - How many repetitions of samples analysis were done?
- - The formula in line 103 should be presented in separate line (see guidelines)
- - Methods should be provided with details
- - Paragraph starting in line 143 is illegible, please consider placing the data in a table.
- - Please correct Table 2 headline.
- - Please provide Figure 1. in higher quality/resolution
- - Please correct the citations, e.g. line 209 is “study by [32]” should be “study by Pazzola et al. [32]”
- - Line 280: please add more citations, because you mention about other studies but add only one publication
- - Please put correct citation in references [11] and [33] both posters, but different form of reference
Author Response
Very good article, the topic is important, and the results and developed models are valuable and allow to explore the topic of coagulation and the impact of the presence of microorganisms on this process. Some issues should be improved:
Literature gap is weakly stated. Please develop introduction.
AU: The introduction has been modified according to the suggestion.
Line 37: …on the production of high-quality…
AU: This sentence has been removed in the new version of the manuscript.
Line 50-51: Please redraft the sentence about the coagulation process
AU: This sentence has been removed in the new version of the manuscript.
How many sheep were on the farms, please provide information about the flocks size and milk yield in the materials and methods section
AU: The following paragraph has been included in accordance with the suggestion: “The average flock size was 1,144 ewes, ranging from 150 to 5,500 ewes, with an average production of 143 kg/ewe/year, varying from 91 to 200 kg, over a mean lactation period of 121 days, ranging from 100 to 150 days. All farms employ mechanical milking systems, and ewes are milked twice daily. The milk is then filtered and stored in refrigerated bulk tanks, maintaining a maximum temperature of 6 °C. The production system is semi-intensive, traditionally associated with grazing on natural pastures, and utilizing residues and remains of cereal crops. On average, each sheep has access to 1.44 ha, ranging from 0.5 to 1.94 ha.”
How many repetitions of samples analysis were done?
AU: As indicated in the methodology, one sample from each farm was analyzed per season.
The formula in line 103 should be presented in separate line (see guidelines)
AU: The manuscript has been modified according to the suggestion.
Methods should be provided with details
AU: The following paragraph has been included in accordance with the suggestion: “Serial dilutions were performed from each sample to inoculate 0.1 mL onto different culture media. Bacterial counts for the following groups of microorganisms were determined on PCA media (Panreac, Spain): standard plate count (SPC) was incubated in aerobic conditions at 30 °C for 72 h [15]; thermoduric bacteria (THERMO) were incubated under the same conditions as SPC, after pasteurizing milk at 62.8 °C for 30 min [16]; psychrotrophic bacteria (PSYCHRO) were incubated at 6.5 °C for 10 days [17]. Pseudomonas spp. (PSEUDO) was cultured on Cetrimide agar (Panreac, Spain) and incubated at 35 °C for 48 h [18]. Determination of Escherichia coli (ECOLI) and other coliforms (COLI) was conducted using CromoIDTM Coli medium (bioMérieux, Spain), incubated at 37 °C for 24 h [19]. Gram-positive catalase-negative cocci count (GPCNC) was determined in modified Edwards medium with colistin and oxolinic acid supplement (Oxoid, UK), incubated at 35 °C for 48 h [9]. Lactic acid bacteria (LAB) were seeded on MRS medium (Panreac, Spain) acidified to pH 5.7, and incubated at 30 °C for 72 h [20]. Lactate-fermenting Clostridium spores (BAB) count was performed using the most probable number (MPN) technique, in Bryant and Burkey Broth (BBB, Merck, Germany) [21]. For the enumeration of coagulase-positive staphylococci (CPS) and coagulase-negative staphylococci (CNS), Baird Parker RPF Agar medium (bioMérieux, Spain) was used, and incubated at 37 °C for 24 h [4]. The microbial counts were expressed in colony-forming units per mL of milk (CFU/mL) and Clostridium spores were expressed as spores/mL. A logarithmic transformation (log10) was applied to both microbial counts to normalize their distribution”.
Paragraph starting in line 143 is illegible, please consider placing the data in a table.
AU: A table with this information has been included as the reviewer's suggestion.
Please correct Table 2 headline.
AU: The headline has been modified according to the suggestion.
Please provide Figure 1. in higher quality/resolution
AU: According to both reviewers, high-quality figures have been provided where equations and coefficients of determination have been incorporated.
Please correct the citations, e.g. line 209 is “study by [32]” should be “study by Pazzola et al. [32]”
AU: We have changed the references. However, we have taken this suggestion into account throughout the manuscript.
Line 280: please add more citations, because you mention about other studies but add only one publication
AU: More references have been included according to the reviewer's suggestion.
Please put correct citation in references [11] and [33] both posters, but different form of reference
AU: Both references has been modified according to the suggestion.

Reviewer 2 Report
Comments and Suggestions for Authors
The presented work includes very interesting and extensive research carried out on 77 sheep farms, with the goal to reach a better understanding of the microbiological factors influencing milk coagulation, providing valuable information for quality control and dairy production management.
​Regarding the presented article, I have several comments and questions that should be explained and supplemented in the article.
- The title of the article states that it is focused on Associations between Technological Characteristics and Microbiological Quality in Sheep Bulk Tank Milk. Therefore, some additional comments are focused in particular on the need to more accurately characterize breeding technology and the milking process.
- Keywords need to be added, perhaps it would be appropriate to arrange them alphabetically.
- Introduction – Of the 43 cited publications, 16 are self-citations, which seems to me to be a relatively large number. It would be appropriate to point more clearly to other theoretical starting points of the given issue and publications by other authors.
- in Methods, it is advisable to state what the average size of the farms was, i.e. the number of dairy ewes from which the milk was examined.
- Among other things, milking equipment and treatment of milk after milking also affect the quality of milk. In Methods, it is necessary to state what the milking equipment was, what treatment of milk after milking and how often the milk was taken to the dairy company for milk processing.
- 3. Results and Discussion are confusing. In Results and Discussion, indicate the equation of the trend line in the graphs and always indicate the reliability R-value, i.e. practically in Figures 1, 2, 3, 4 for I. and II. tightness of dependence around the fitted linear dependence. The points are very scattered in the images, how tight is the dependence around the interpolated lines?
- Conclusions – it is necessary to modify and supplement with specific results resulting from the conducted research, e.g. instead of the general phrase: "A significant association was found between seasonal variations (Season) and all coagulation properties..." it is necessary to state the specific results, which time of year manifested, as manifested, etc. also in other research results. This specific information about the research results should also be added to the Abstract.
Author Response
The presented work includes very interesting and extensive research carried out on 77 sheep farms, with the goal to reach a better understanding of the microbiological factors influencing milk coagulation, providing valuable information for quality control and dairy production management.
​Regarding the presented article, I have several comments and questions that should be explained and supplemented in the article.
The title of the article states that it is focused on Associations between Technological Characteristics and Microbiological Quality in Sheep Bulk Tank Milk. Therefore, some additional comments are focused in particular on the need to more accurately characterize breeding technology and the milking process.
AU: We appreciate the suggestion. To avoid confusion, we have changed the article's title. We now believe it better reflects the article's content: "Associations between milk coagulation properties and microbiological quality in sheep bulk tank milk".
Keywords need to be added, perhaps it would be appropriate to arrange them alphabetically.
AU: We have rearranged and added more keywords as suggested.
Introduction – Of the 43 cited publications, 16 are self-citations, which seems to me to be a relatively large number. It would be appropriate to point more clearly to other theoretical starting points of the given issue and publications by other authors.
AU: The introduction has been modified taking into account this and other suggestions from reviewer 1. Now the paper has 41 cited with 6 self-citations (less than 15%).
In Methods, it is advisable to state what the average size of the farms was, i.e. the number of dairy ewes from which the milk was examined.
AU: This section has been modified taking into account this and other suggestions from reviewer 1.
Among other things, milking equipment and treatment of milk after milking also affect the quality of milk. In Methods, it is necessary to state what the milking equipment was, what treatment of milk after milking and how often the milk was taken to the dairy company for milk processing.
AU: This section has been modified taking into account this and other suggestions from reviewer 1.
- Results and Discussion are confusing. In Results and Discussion, indicate the equation of the trend line in the graphs and always indicate the reliability R-value, i.e. practically in Figures 1, 2, 3, 4 for I. and II. tightness of dependence around the fitted linear dependence. The points are very scattered in the images, how tight is the dependence around the interpolated lines?
AU: According to both reviewers, high-quality figures have been provided where equations and coefficients of determination have been incorporated.
Regarding the adjustment of trend lines, we must say that the milk composition variables (fat, lactose, caseins, and pH) have been included in the mixed linear models as covariate factors. For this reason, the coefficients of determination are low. However, this does not mean that the covariate factor is not a significant predictor of the coagulation variables; in fact, the F and P values in Table 3 indicate a significant association. What we are observing is more of a trend than a precise mathematical or statistical dependency.
Conclusions – it is necessary to modify and supplement with specific results resulting from the conducted research, e.g. instead of the general phrase: "A significant association was found between seasonal variations (Season) and all coagulation properties..." it is necessary to state the specific results, which time of year manifested, as manifested, etc. also in other research results. This specific information about the research results should also be added to the Abstract.
AU: The abstract and conclusions have been modified following the suggestions of the reviewer.

Round 2
Reviewer 2 Report
Comments and Suggestions for Authors
The article was partially edited according to the reviewer's instructions and the authors' possibilities. I recommend it for publication.